# Transcriptome Comparison Reveals Key Components of Nuptial Plumage Coloration in Crested Ibis

**DOI:** 10.3390/biom10060905

**Published:** 2020-06-15

**Authors:** Li Sun, Tong Zhou, Qiu-Hong Wan, Sheng-Guo Fang

**Affiliations:** MOE Key Laboratory of Biosystems Homeostasis & Protection, State Conservation Centre for Gene Resources of Endangered Wildlife, College of Life Sciences, Zhejiang University, Hangzhou 310058, China; sunli2012@zju.edu.cn (L.S.); zhoutong2016@zju.edu.cn (T.Z.); qiuhongwan@zju.edu.cn (Q.-H.W.)

**Keywords:** crested ibis, feather regeneration, melanin, skin transcriptome

## Abstract

Nuptial plumage coloration is critical in the mating choice of the crested ibis. This species has a characteristic nuptial plumage that develops from the application of a black sticky substance, secreted by a patch of skin in the throat and neck region. We aimed to identify the genes regulating its coloring, by comparing skin transcriptomes between ibises during the breeding and nonbreeding seasons. In breeding season skins, key eumelanin synthesis genes, *TYR*, *DCT*, and *TYRP1* were upregulated. Tyrosine metabolism, which is closely related to melanin synthesis, was also upregulated, as were transporter proteins belonging to multiple *SLC* families, which might act during melanosome transportation to keratinocytes. These results indicate that eumelanin is likely an important component of the black substance. In addition, we observed upregulation in lipid metabolism in breeding season skins. We suggest that the lipids contribute to an oil base, which imbues the black substance with water insolubility and enhances its adhesion to feather surfaces. In nonbreeding season skins, we observed upregulation in cell adhesion molecules, which play critical roles in cell interactions. A number of molecules involved in innervation and angiogenesis were upregulated, indicating an ongoing expansion of nerves and blood vessels in sampled skins. Feather β keratin, a basic component of avian feather filament, was also upregulated. These results are consistent with feather regeneration in the black skin of nonbreeding season ibises. Our results provide the first molecular evidence indicating that eumelanin is the key component of ibis coloration.

## 1. Introduction

Crested ibis (*Nipponia nippon*) wear a lead grey nuptial plumage (Figure 1), which is a critical sign of genetic quality in the competition for mates [1]. The components of the black cosmetic material used in plumage coloration have not yet been identified. Since the “secretion” of this unique material has only been found in crested ibis, we believe that clarifying the composition of this black substance will contribute to an understanding of avian pigment application. Moreover, considering the effect of nuptial plumage on mate choice, information on the constituents of the black substance will finally benefit the conservation of this endangered species.

Birds are masters in using structural or pigmentary color to arm themselves, making them the most colorful terrestrial vertebrates [2]. Bright, expressive colors are often used in nuptial plumage. Blue and green colors can be formed by feather nanostructure, as in painted buntings (*Passerina ciris*), while red, orange, and yellow can be formed by carotenoids, as in painted buntings (*Passerina ciris*) and male American goldfinches (*Spinus tristis*) [3]. Less common pigments used by birds include the psittacofulvins found in parrot feathers [4]. The combination of structural and pigment color enriches avian ornamental coloration, such as the extensively diverse plumage color in fairy wrens (*Malurus* spp.) [5]. In wild budgerigar (*Melopsittacus undulatus*), the green plumage coloration forms by combining a blue structural color and a yellow pigmentary (psittacofulvins) color. Substitution of a single amino acid could abolish the activity of a specific polyketide synthase, which might catalyze psittacofulvin synthesis, and cause a blue instead of green plumage color in budgerigars [6]. Melanin, the most common pigment used by birds [7], seems to play a relatively subordinate role in nuptial plumage coloration. Feather melanin includes eumelanin and pheomelanin, with eumelanin yielding brown and black colors, and pheomelanin yielding yellow and red colors [8], which are duller (to human eyes) than structural and carotenoid colors.

Crested ibis follow unique morphological, physiological, and behavioral changes to develop a lead grey nuptial plumage coloration [9]. Before the breeding season starts, a patch of black skin on the throat and neck region of adult crested ibis begins to “secrete” a tar-like black substance, which is transferred to feathers on the back, head, and neck through diligent bathing and daubing behaviors. Interestingly, prior to the color “secretion”, adult crested ibises experience a localized feather regeneration in the black neck skin region. Older downy feathers are replaced by a specialized downy feather with long rachises and short lateral barbs, which may reduce the feather surface to which the black substance can adhere, and this will enable more efficient transfer during subsequent bathing and daubing [9].

The accurate chemical composition of ibis black pigment has not yet been determined, although melanin is probably an important component [9]. In a previous study, we found that lower ultraviolet reflectance of the black substance reflects a higher genetic quality, and could work as a visual cue for female crested ibises to choose mates [1]. In the wild, the dark plumage coloration could help parents conceal themselves and protect their babies from predators when nesting. As such, the darker pigment may correlate with reproductive success. Different components might influence the overall reflectance of the substance, but few studies have examined its composition. It is poorly soluble in water and in most organic solvents, so chromatography or mass spectrometry approaches are limited [10]. However, there are polygonal melanocyte-like cells underneath the epidermis of the black skin [11], and their presence supports the notion of melanin contribution. In this study, we compared the black pigment-producing skin in nonbreeding season molting ibises with that of breeding season secreting ibises, using high-throughput mRNA-seq data, with the aim of identifying putative genes or pathways that may be related to the formation and application of the black substance.

## 2. Materials and Methods

### 2.1. Sampling and RNA Extraction

Skin samples were collected from two different crested ibises during nonbreeding (November 2017) and breeding (April 2018) seasons at the breeding base in Tashui, Shawan District, Chengdu City. Skin tissue was sheared from the black neck skin of each bird and immediately stored in liquid nitrogen, before subsequent extraction. We used a scalpel to cut a small patch of skin (approximately 1 cm^2^) at the sampling location, and the depth of the cut reached the dermis. Then, we peeled off this small piece of skin from the sampled individual. The sampled tissue includes the epidermis, dermis, and all structures of the feather follicle, including the dermal papilla and the lower part of the feather. We used a TRIzol total RNA isolation kit (Invitrogen, Carlsbad, CA, USA) to extract total RNA, according to the manufacturer’s protocol. The RNA integrity number (RIN) of all four samples was calculated (Appendix A). We used RIN values of 6.8 and higher for library construction.

### 2.2. cDNA Library Construction and Sequencing

For each sample, we purified mRNA from an amount of 1 μg total RNA using poly-T oligo-attached magnetic beads and constructed cDNA libraries using NEBNext^®^ Ultra™ Directional RNA Library Prep Kit for Illumina^®^ (NEB, Ipswich, MA, USA), according to the manufacturer’s protocol. We used the Agilent Bioanalyzer 2100 system to assess the quality of our cDNA libraries. Index-coded sample clustering was performed on a cBot Cluster Generation System using TruSeq PE Cluster Kit v3-cBot-HS (Illumina), following the manufacturer’s recommendations. Then cDNA libraries were sequenced on an Illumina Hiseq platform, and paired-end reads were generated.

### 2.3. Data Analysis

Quality control. Raw reads were processed to remove reads containing adapter, poly-N, and low-quality reads. We calculated the Q20, Q30, and GC content of the clean reads. Subsequent analyses were based on the highly qualified clean reads.

Mapping. A reference genome of the crested ibis has been uploaded to GenBank (accession: GCA_000708225) [12]. We downloaded the genome and gene model annotation files and used Hisat2 v2.0.4 to build the index of the reference genome and align paired-end clean reads to the reference genome.

Novel genes prediction. Cufflinks was used to assemble mapping results. Cuffcompare was used to compare the current genes with previously published gene sets, to find novel genes that have not been annotated, to find new exons in known genes, and to optimize the start and end position of known genes. A new set of annotated genes, built with the previous set and novel genes, was used in subsequent analysis.

Gene expression quantification. We used the HTSeq v0.9.1 software to count the numbers of reads mapped to each gene, then used fragments per kilobase of transcript sequence per million base pairs sequenced (FPKM) to estimate the gene expression level [13]. The FPKM of each gene was calculated, based on gene length and read counts mapped to the gene. Genes with FPKM > 1 were regarded as actively expressed genes.

Differential expression analysis. We used the DESeq R package (1.18.0) to analyze differential expression between the two groups. DESeq provides statistical routines for determining differentially expressed genes, using a negative binomial distributed model. Adjusted *p*-values (*q*) were obtained by adjusting original *p*-values using Benjamini and Hochberg’s approach to control the false discovery rate. Genes with adjusted *p* < 0.05 were assigned as differentially expressed genes (DEGs).

Gene Ontology (GO) term and Kyoto Encyclopedia of Genes and Genomes (KEGG) pathway analyses. GO enrichment analysis of DEGs was conducted using the GOseq R package, to correct the bias of gene length. We considered a GO term to be significantly enriched by DEGs at adjusted *p* < 0.05. KOBAS was used to test the statistical enrichment of DEGs in KEGG pathways. KEGG pathways with adjusted *p*-values < 0.05 were considered significantly enriched by DEGs.

## 3. Results

### 3.1. Sequencing Data Summary

A total of 205,130,880 paired-end reads were produced from the sequencing, among which 95.08% (195,045,578) met quality control standards. Among these, 77.14% (150,460,359) were mapped to the reference genome sequence (Table 1). We identified 2359 novel protein-coding genes (Appendix A), which were added to the previously annotated set of 17,076 protein-coding genes. Subsequent analyses were based on the new gene set of 19,435 genes.

### 3.2. Differential Gene Expression Analysis

In the new gene set, 14,808 genes were expressed actively in at least one of the four samples. Among these, 2055 belonged to novel genes identified here (Appendix A). Differential gene expression analysis showed that there were 518 DEGs between breeding season and nonbreeding season skins. In breeding season ibises, 136 DEGs were upregulated, and 382 were downregulated (Figure 2, Appendix A). Table 2 gives partial DEGs involved in several categories, which might play roles in the black skin region of nonbreeding season and breeding season ibises.

### 3.3. Breeding Season

GO terms and KEGG pathways enriched during the breeding season are shown in Appendix A and Table 3, respectively. All GO terms including phospholipid catabolic processes (*q* = 3.44 × 10^−3^), phospholipase activity (*q* = 8.06 × 10^−3^), lipase activity (*q* = 1.12 × 10^−2^), and cellular lipid catabolic process (*q* = 1.14 × 10^−2^), as well as four of five upregulated K × 10GG pathways including α−linolenic acid metabolism (*q* = 2.26 × 10^−2^), arachidonic acid metabolism (*q* = 2.26 × 10^−2^), linoleic acid metabolism (*q* = 2.26 × 10^−2^), and ether lipid metabolism (*q* = 4.11 × 10^−2^) fell under the category of lipid metabolism. Tyrosine metabolism, which is closely related to melanogenesis, was also upregulated (*q* = 2.38 × 10^−2^; Table 2). Three key melanin synthesis genes were significantly upregulated, including tyrosinase (*TYR*; *q* = 7.19 × 10^−5^), which catalyzes tyrosine to dopaquinone (*DQ*); dopachrome tautomerase *(DCT*; *q* = 2.36 × 10^−6^), which isomerizes dopachrome to 5,6−dihydroxyindole−2−carboxylic acid (DHICA); and tyrosinase−related protein 1 (*TYRP1*; *q* = 6.08 × 10^−4^), which increases the production of eumelanin (Figure 3). Finally, several members of the solute carrier family were upregulated, suggesting transmembrane movement after eumelanin synthesis (Table 2).

### 3.4. Nonbreeding Season

Nonbreeding season samples showed upregulation of cell adhesion molecules, their ECM ligands, and cytoskeleton proteins (Table 2). KEGG analysis showed four enriched pathways (Table 3), three of which were ECM-receptor interaction (*q* = 3.71 × 10^−10^), focal adhesion (*q* = 1.52 × 10^−9^), and cell adhesion molecules (CAMs; *q* = 1.19 × 10^−3^), all related to cell interactions. GO analysis also showed enrichment in GO terms, including extracellular matrix, collagen trimer, biological adhesion, and cell adhesion (Appendix A). These results indicate strong interactions between intracellular and extracellular environments, potentially due to requirements for cell proliferation and migration.

Upregulated DEGs were also involved in innervation molecules, tumorigenesis, and angiogenesis, indicating rapid cell proliferation due to nerve and blood vessel expansion (Table 2). Consistently, upregulations were found in receptors of several cell growth factors, regulators of the cell cycle, and *NOTCH2*, which can regulate cell fate. Further, several morphogens and the matrix metalloproteinases *MMP2* and *MMP16* were upregulated in nonbreeding season samples. *MMP2* and *MMP16* play roles in epithelium invagination and mesenchymal cell proliferation during chicken feather follicle formation, indicating tissue morphogenesis and remodeling occurring in nonbreeding season samples.

The phagosome KEGG pathway was also upregulated (*q* = 3.49 × 10^−2^; Table 3), suggesting the involvement of innate immunity and tissue remodeling. GO analysis showed enrichments in two adaptive immunity terms, including the MHC protein complex (*q* = 1.68 × 10^−3^) and MHC class II protein complex (*q* = 1.68 × 10^−3^; Appendix A). Immune genes, including MHC class II antigens, macrophage mannose receptors, lymphocyte proteins, and complement molecules, were upregulated during the nonbreeding season (Table 2).

## 4. Discussion

Our results indicate high eumelanin synthesis in sampled skins in the breeding season, while active cell proliferation and tissue remodeling take place in the nonbreeding season.

### 4.1. Melanin Synthesis and Transport

Melanin is the most common used pigment in vertebrates, and is responsible for the dark coloration of skin, feather, fur, eyes and connective tissues [7]. Genetic basis under melanin synthesis has been extensively studied. *TYR* is a rate-limiting enzyme in melanocytes. In tyrosine metabolism, tyrosine is catalyzed by *TYR* to form *DQ*, a common precursor of eumelanin and phaeomelanin. Although there was reported that *TYR*-independent mutation, such as mutations in EDNRB2 [14] and NDP [15], can cause different plumage pigment pattern, abnormal function of *TYR* could lead to an inability to normally synthesize melanin, resulting in white phenotypes [16]. In White Plymouth Rock chicken, an avian retroviral sequence insertion in *TYR* intron 4 leads to substantially reduction in normal transcript and results white plumage [17]. The upregulation of *TYR* could promote the overall rate of melanin biosynthesis. In the absence of sulfhydryl compounds (e.g., cysteine), *DQ* will undergo intramolecular cyclization, generating cyclodopa, which will be immediately oxidized into more stable dopachrome. *DCT* isomerizes dopachrome to generate DHICA [18], which further forms eumelanin polymers under the action of *TYRP1*. *TYRP1* also maintains *TYR* stability, regulates *TYR* catalytic activity, and plays a part in melanocyte maturation, proliferation, and apoptosis [19,20]. In previous studies, black animals showed higher levels of *TYR*, *TYRP1*, and *DCT* than white animals [21,22,23]. Our results showed that, during the production of the black substance in the breeding season, *TYR*, *TYRP1*, and *DCT* were upregulated in black skin, indicating that eumelanin is likely an important component of the black substance.

After synthesis, eumelanin is transported to keratinocytes nearer to the skin surface. Melanin-containing mature melanosomes transfer into adjacent keratinocytes through melanocyte dendrite tips [24]. Various transmembrane proteins are involved in this process. We found upregulation of multiple members of the solute carrier family, including *SLC45A2*, a transporter protein that has been proved to mediate melanin synthesis. Mutations in *SLC45A2* cause light skin coloration in humans, horses, tigers, and dogs [25,26,27].

The putative process of the production and discharging of the eumelanin is illustrated in Figure 4. We observed that during breeding, eumelanin synthesis is greatly improved, leading to increasing pigment deposition in the black skin region. It is possible that these colored keratinocytes rapidly become cuticles and are shed from the skin surface, forming an important color-determining component of the black substance.

### 4.2. Lipids

The black substance has been suggested to contain a water-insoluble oil base [9], which is supported by our finding of upregulated lipid metabolism pathways in breeding season black skin. Therefore, we suggest lipids as another important component of the black substance.

A widely studied avian secretion is uropygial gland secretion, which is also composed of various lipids. Uropygial secreted lipids have been reported to show pheromone activity in female birds, enhance courtship plumage, resist parasites, and repel water [28]. In greater flamingoes (*Phoenicopterus roseus*), the uropygial secretion additionally contains pigments, such as basic carotenoids [29]. Similar to the daubing behavior of breeding crested ibis, greater flamingoes work harder during the breeding season, to spread uropygial secretions over their feathers to enhance their pink coloration, indicating that these secretions have roles in mate choice of both birds.

The cosmetic effect of secreted pigments in flamingoes and crested ibises could be promoted by the accompanying lipids, since lipids can add viscidity to the secretion, and enhance its adhesion to feather surface during daubing behavior, giving gloss to the plumage and making it more brilliant. In addition, lipid metabolism upregulation might be due to increased energy requirements during the breeding season. Lipid metabolism can break down stored fats to provide energy for reproduction. In mice, impaired lipid metabolism can cause infertility [30]. Birds make an enormous reproductive investment: males attract females by changing their appearance, occupying sites, and providing food, and females must lay nutrient-containing eggs. For crested ibis, parents have to care for their offspring for at least 40 days after hatching [31]. Thus, ibises require considerable energy from lipids in the breeding season.

Most DEGs enriched in lipid metabolism were phospholipase A2 (PLA2s), which encode rate-limiting enzymes in the synthesis of arachidonic acid, which stimulates basal progesterone and androstenedione production [32]. Studies in quails have shown that PLA2 activity can be regulated by estradiol [33], suggesting that the upregulation of PLA2s indicates higher sex hormone levels in breeding season crested ibis skins. Estradiol levels in crested ibis are positively correlated with the presence of bathing and daubing behavior [11], and the amount of secreted black coloration [10]. Thus, the upregulation of PLA2s in breeding season ibises could regulate the formation of the nuptial plumage.

In future studies, some improvements can be made to increase research robustness. Transcriptome profiling have shown that feathers from different parts of the body, characterized by different forms and serving different functions, undergo different genetic bases during generation at embryonic stages or regeneration thereafter [34,35,36], thus comparison between the black skin and other skin regions could help infer function of differentially expressed genes. Proteomic profiling could provide further evidences of the genetic differences underlying formation of different feathers [37]. On the other hand, the transcriptomic comparison of other skins between breeding and non-breeding season could reduce the difference caused by other physiological changes, instead of production of the black substance, which increase the causality of research results. For example, it will be clearer whether lipid metabolism upregulation is due to uropygial secretion in the breeding season like other birds, or due to the production of the black substance uniquely in crested ibis. Furthermore, pigment pattern differences between the black skin and other skin regions could be tracked to the establish of melanocyte progenitor distribution at early embryonic stage [38]. It would be interesting to uncover the genetic basis underlying the formation of the skin pigment pattern along the embryonic time course.

### 4.3. Feather Regeneration

In the nonbreeding season, when the skin samples were collected, prior to the secretion of nuptial plumage cosmetic material, the adult crested ibises showed localized feather regeneration in the black skin region. Feather regeneration is an integrated process of cell proliferation, migration, differentiation, and apoptosis, requiring considerable expression of molecules involved in cell growth, cell division, and cell migration. Various types of morphological changes have to occur to build a new feather, demanding dynamically changing genetic basis for different feather structures [35,39]. In addition, the formation of a new feather filament requires the biosynthesis of component materials, such as feather keratin.

Feather regeneration is initiated by a structure called the dermal papilla (DP), localized at the base of the feather follicle (Figure 5). The upregulation of *NCAM1*, a DP biomarker, indicated that feather growth had begun in nonbreeding season samples. Another DP marker, tenascin (TNC), showed a seven-fold increase in expression in the nonbreeding season, compared to the breeding season. We also observed upregulation in a tenascin-X-like gene, which might have a similar function. Through an unknown signaling mechanism, DP induces stem cells (STCs) on the epidermal collar to proliferate [40]. STCs proliferate throughout the feather growth phase, differentiating into keratinocytes and supportive cells [40]. Cell growth factor receptors, including *EGFR*, *PDGFRs*, and *TGFRs*, were upregulated in our results, and probably play important roles in STC proliferation. Interestingly, we also observed upregulation in the proto-oncogene SKI, which, when overexpressed, has been shown to promote cell proliferation in tumors [41]. Multipotential STC progenies move into a multilayered ramogenic zone (RZ) epithelium, which invaginates and forms barb ridges (BRs) [40]. The migration of STC progenies demands cell-cell and cell-extracellular matrix (ECM) interactions [42], which explains the considerable upregulation of CAMs we observed. Cell differentiation happens as anterior BRs fuse to develop feather rachis, while posterior BRs separate and develop into feather branches. The upregulation of *NOTCH2*, which is involved in determining the fate of various cells [43,44,45], might play a role in the dictation of the STCs. Posterior BRs differentiated into barbule plates (BP), which keratinize into feather branches, and marginal plates (MP) and axial plates (AP), which degenerate to form interspace separating adjacent branches. Keratinization occurs in the cytoplasm of keratinocytes, which further develop into feather bone structure, including rachis, barbules, and branches. Keratinocytes produce intermediate filament keratin (IFK), most of which is replaced by feather beta keratin (FbetaK) at keratinization. Keratin-associated proteins (KAP) cross-link the keratin filaments into large bundles of corneous filament materials [46]. Upregulation in FbetaK, IFKs, and KAPs in nonbreeding season skins indicates ongoing BP keratinization in some skin follicles. Following BP keratinization, MP and AP degenerate through cell apoptosis. In chickens, MP apoptosis is induced by SHH and inhibited by BMP2 and BMP4 [47,48]. We observed upregulation in SHH and NIBAN1, indicating that MP cells in some follicles were degenerating normally.

As STC progeny migrate towards the distal end of the follicle, a cylindrical center forms, and this is filled with a pulp of loosely packed mesenchymal cells generated from a DP rich in ECM. This explains the upregulation of ECM molecules, such as collagens, filamin, and laminin. We found that a number of neural development factors (*NEGR1*, *CDH11*, *SDK1*, *SDK2*, *FNT1*, *SEMAs*, *NRPs*, *NTN4*) and angiogenic factors (*SEMA3s*, *MMPs*) were upregulated in nonbreeding season skins, indicating the formation of new nerves and blood vessels during feather growth. Studies of chicken feather regeneration have indicated that feather follicles are vascularized and innervated, so that newly grown feathers are integrated with adjacent tissues, and nutrition and signals can be provided for feather formation via these connections [40]. During nerve formation, *SEMAs* and *NTN4* act as attractants or repellents to help neurons to find precise paths, by which they will send out axons [49]. The CDHs and SDKs expressed on the glial cells or neurons, and the fibronectin and laminin in ECM physically interact with axon protrusions, forming synapses between two neurons [50,51]. Neurexin 1 (*NRXN1*), a presynaptic protein that connects neurons at synapses, was also upregulated. In addition, the roles of *SEMAs* in branching morphogenesis of vascular system, lung, and kidney [52,53] indicate that they might also play a role in angiogenesis and feather branching. *MMPs* have been reported to promote new capillary formation by deconstructing ECM, which could decrease the integrity of the microvasculature [54,55]. 

### 4.4. Immunity

Nonbreeding season molting skins had higher levels of multiple MHC class II antigens and their trans-activator (CIITA), and showed higher adaptive immune responses than breeding season secreting skins. We speculate that there are two reasons for the different adaptive immune levels. First, winter (when the non-breeding season samples were collected) is a season of high incidence of viral diseases, such as bird flu, which may challenge the immune system of the crested ibis. Second, in breeding season, there may be a resource-allocation trade-off between immune investment and breeding activity investment. In pied flycatchers, the artificially up-regulated immune activity of males who provisioned nestlings led to delayed post-nuptial molt [56], indicating a resource-competing relationship between immunity, reproduction and molt. A research of burying beetles showed that immune suppression occurs during parental care [57]. Crested ibis provide considerate parental care for their young, which might cost so much energy that they have to compromise immune level. We observed that the upregulated immune genes were involved in the Influenza A pathway, warning of potential avian influenza in winter molting crested ibis. Macrophage markers *MPEG1*, *MRC2*, and CX3CR1 might upregulate for antigen presentation to activate MHC genes. In addition, a recent study reported that upon skin wounding, dermal macrophages could contribute to localized nerve regeneration [58]. Therefore, upregulations in macrophage markers might also contribute to the innervation during feather regeneration.

## Figures and Tables

**Figure 1 biomolecules-10-00905-f001:**
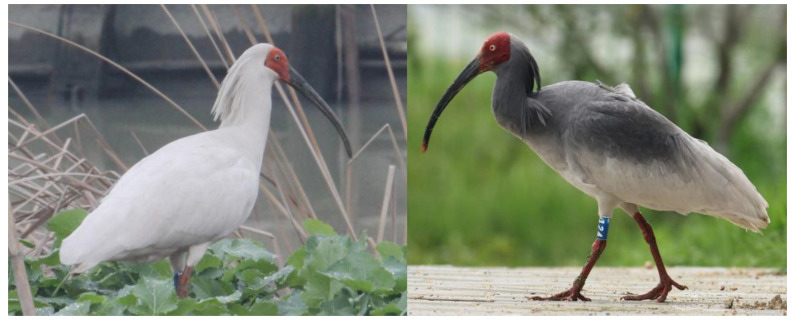
Plumage of adult nonbreeding season ibis (**left**) and adult breeding season ibis (**right**).

**Figure 2 biomolecules-10-00905-f002:**
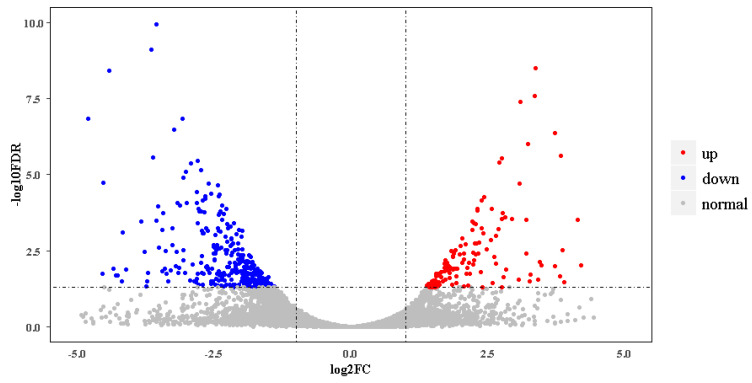
Volcano plot of differentially expressed genes (DEGs) of black skins between breeding and nonbreeding season ibises. Upregulated and downregulated DEGs are shown as red and blue dots, respectively. The label of the *x*-coordinate log2FC is the log2 value of the fold change (FC) between breeding season ibises and nonbreeding season ibises. The label of the *y*-coordinate‒log10FDR is the log10 value of the adjusted *p*-value.

**Figure 3 biomolecules-10-00905-f003:**
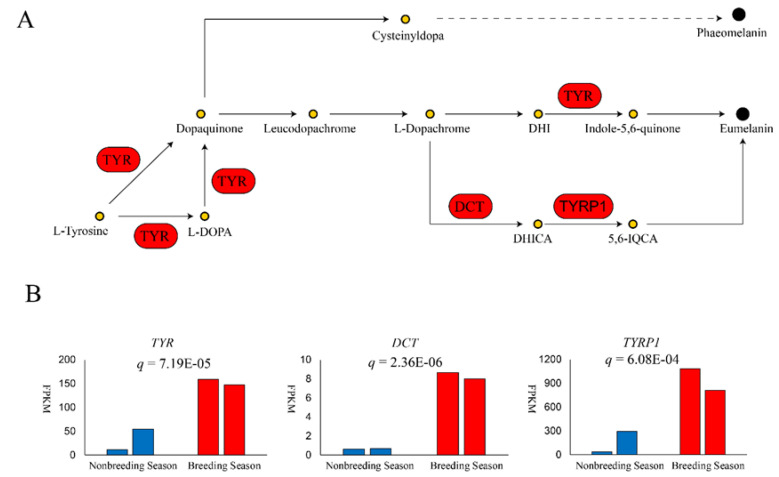
Upregulation of key genes in synthesis of eumelanin in breeding season. (**A**) shows the process of eumelanin and pheomelanin synthesis. (**B**) gives a comparison of expression of three genes *TYR*, *DCT*, and *TYRP1* in eumelanin synthesis. The vertical axis of three bar graphs represents the fragments per kilobase of transcript sequence per million base pairs (FPKM) sequenced of each gene. Adjusted *p* values are shown as *q* in each bar graph.

**Figure 4 biomolecules-10-00905-f004:**
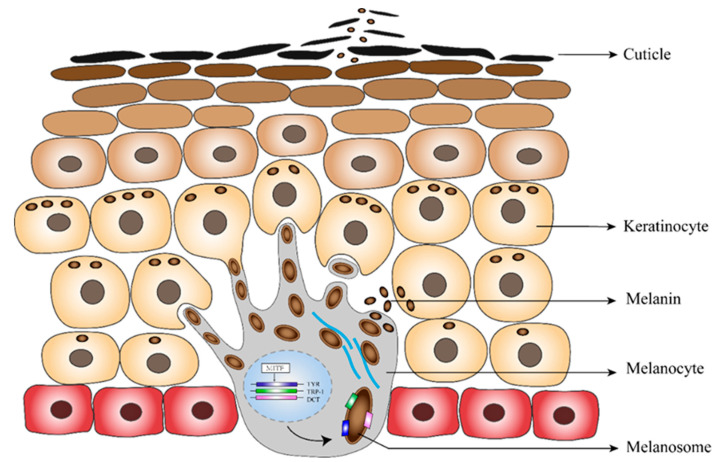
Putative formation of the black skin secretion.

**Figure 5 biomolecules-10-00905-f005:**
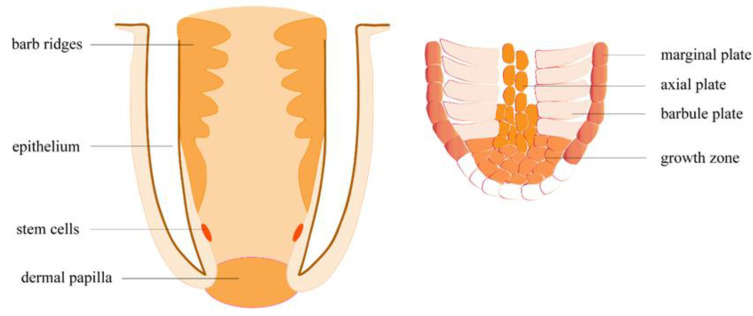
Follicle structure during feather regeneration.

**Table 1 biomolecules-10-00905-t001:** Summary of mRNA-seq data.

	WF1	WF2	BF1	BF2
Total reads	52,113,900	52,658,080	43,781,878	46,491,720
Total mapped	40,559,579 (77.83%)	40,863,473 (77.6%)	32,666,772 (74.61%)	36,370,535 (78.23%)
Multiple mapped	2,237,161 (4.29%)	2,340,580 (4.44%)	2,164,084 (4.94%)	2,014,853 (4.33%)
Uniquely mapped	38,322,418 (73.54%)	38,522,893 (73.16%)	30,502,688 (69.67%)	34,355,682 (73.9%)
Reads map to ‘+’	19,179,652 (36.8%)	19,267,372 (36.59%)	15,273,761 (34.89%)	17,190,529 (36.98%)
Reads map to ‘−’	19,142,766 (36.73%)	19,255,521 (36.57%)	15,228,927 (34.78%)	17,165,153 (36.92%)
Non-splice reads	24,473,684 (46.96%)	25,126,246 (47.72%)	21,155,595 (48.32%)	22,077,856 (47.49%)
Splice reads	13,848,734 (26.57%)	13,396,647 (25.44%)	9,347,093 (21.35%)	12,277,826 (26.41%)

WF (white feather) represents skin samples from nonbreeding season ibises and BF (black feather) represents skin samples from breeding season ibises.

**Table 2 biomolecules-10-00905-t002:** Partial upregulated DEGs in breeding and nonbreeding season samples.

Stage	Category	Representative Molecules
Breeding Season	Melanogenesis	*TYR, DCT, TYRP1*
	Lipid metabolism	*PLA2G4EL2, PLA2G4EL3, PLA2G4EL4, PLA2G4EL7*
	Transporter proteins	*SLC6A9, SLC24A4, SLC25A17, SLC34A2, SLC38A4, SLC45A2*
Nonbreeding Season	Cell adhesion	*ITGA1, ITGA9, ITGA11, ITGBL1, CDH11, COL1A1, COL1A2, COL3A1, COL4A1, COL5A1, COL5A2, COL6A1, COL6A2, COL6A3, COL6A6, COL8A2, COL12A1, COL15A1, COL17A1, FN1, LAMC3*
	Cell growth, division and differentiation	*PDGFRA, PDGFRB,* *EGFR, TGFBR2, TGFBR3,* *RGCC, NOTCH2*
	Innervation	*NRP1, NRP2,* *SEMA3A, SEMA3A, SEMA3A, SEMA5A, SEMA5B,* *SDK1, SDK2,* *NRXN1,* *NCAM1*
	Tumorigenesis and Angiogenesis	*SKI, NRPs, NTN4-like, NAN2,* *CERCAM*
	Morphogen and tissue rebuilding molecules	*SHH,* *BMP8A-like, WNT11,* *MMP2, MMP16*
	Feather and cytoskeletal keratins	*feather beta keratin, type I cytoskeletal 14 keratin, keratin-associated protein 20-1*
	Immunity	*class II histocompatibility antigen B-L beta chain, HLA class II histocompatibility antigen gamma chain, LY86, C1R*

**Table 3 biomolecules-10-00905-t003:** Enriched KEGG pathways upregulated in black skins of the crested ibises in breeding and nonbreeding seasons.

Terms	Input DEG Number	Background Gene Number	*q*
Upregulated in breeding season			
α-Linolenic acid metabolism	4	25	2.26 × 10^−2^
Arachidonic acid metabolism	5	48	2.26 × 10^−2^
Linoleic acid metabolism	4	28	2.26 × 10^−2^
Tyrosine metabolism	4	31	2.38 × 10^−2^
Ether lipid metabolism	4	39	4.11 × 10^−2^
Upregulated in nonbreeding season			
ECM-receptor interaction	18	73	3.71 × 10^−10^
Focal adhesion	25	184	1.52 × 10^−9^
Cell adhesion molecules	12	112	1.19 × 10^−3^
Phagosome	10	127	3.49 × 10^−2^

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
