# Peer review of "Transcriptome Comparison Reveals Key Components of Nuptial Plumage Coloration in Crested Ibis"

_biomolecules, 2020, doi:10.3390/biom10060905_

Round 1

Reviewer 1 Report

The authors applied the transcriptomic approach to identify genes involved in the pigment synthesis pathways regulating the black skin of crested ibis during breeding seasons. They found some differentially expressed genes and proteins that were enriched on the melanogenesis pathway in the breeding season.

Major comments:

The finding here is minor, and the title is very misleading. The authors used only skin for RNA extraction. Although dermal papilla of feather follicle is supposed to be included, it accounts for only a small portion of the total skin tissue.

Minor Comments:

1) In the Materials and Methods section, the authors stated that a piece of skin was used for RNA extraction. Please clarify that which kind of tissues was used, only the skin with regenerating feather? The epithelium and pulp of feather were also included?

2) Skin is a very complex organ which contains multiple types of tissues, and please provide enough details to clarify how the skin was peeled and how many types of tissues were supposedly included.

3) The melanin can be dissolved in the solution when RNA is extracted. Melanin can inhibit enzymatic reactions that are required for making libraries and sequencing, so please clarify if the final solution of RNA is colorless or not.

4) In Fig. 3B, please clearly indicate labels of x- and y-axis in the figure.

5) The authors did not cite enough previous efforts on studying the mechanism and genetics of avian color differences, and I strongly suggest they should cite and discuss these following papers:

Lin SJ, Foley J, et al. . 2013. Topology of feather melanocyte progenitor niche allows complex pigment patterns to emerge. Science  340(6139):1442–1445.

Smyth JRJr. 1990. Chapter 5. Genetics of plumage, skin and eye pigmentation in chickens. In: Crawford RD, editor. Pultry breeding and genetics.  Amsterdam: Elsevier. p. 109–168.

Sato S, Otake T, Suzuki C, Saburi J, Kobayashi E. 2007. Mapping of the recessive white locus and analysis of the tyrosinase gene in chickens. Poult Sci.  86(10):2126–2133.

Kinoshita K, et al. . 2014. Endothelin receptor B2 (EDNRB2) is responsible for the tyrosinase-independent recessive white (mo(w)) and mottled (mo) plumage phenotypes in the chicken. PLoS One  9(1):e86361.

Cooke TF, et al. . 2017. Genetic mapping and biochemical basis of yellow feather pigmentation in budgerigars. Cell  171(2):427–439 e421.

Vickrey AI, et al. . 2018. Introgression of regulatory alleles and a missense coding mutation drive plumage pattern diversity in the rock pigeon. Elife  7(pii):e34803.

Wang X, et al. Combined Transcriptomics and Proteomics Forecast Analysis for Potential Genes Regulating the Columbian Plumage Color in Chickens. PLoS One. 2019; 14(11): e0210850.

Fan M, et al. Multiple components of feather microstructure contribute to structural plumage colour diversity in fairy-wrens. Biological Journal of the Linnean Society, Volume 128, Issue 3, November 2019, Pages 550–568.

6) The authors did not cite enough previous efforts on transcroptomic analyses of skin appendages, I strongly suggest they should cite and discuss these following papers:

Ng CS, et al. . 2015. Transcriptomic analyses of regenerating adult feathers in chicken. BMC Genomics 16:756.

Gong H, et al. . 2018. Skin transcriptome reveals the dynamic changes in the Wnt pathway during integument morphogenesis of chick embryos. PLoS One  13(1):e0190933.

Li A, et al. . 2017. Diverse feather shape evolution enabled by coupling anisotropic signalling modules with self-organizing branching programme. Nat Commun . 8:ncomms14139.

Yang J, Qu Y, Huang Y, Lei F. 2018. Dynamic transcriptome profiling towards understanding the morphogenesis and development of diverse feather in domestic duck. BMC Genomics  19(1):391.

Reviewer 2 Report

This paper tries to understand the genetic basis for a key trait on Crested Ibis, the nuptial plumage on breeding season. They approach this question by comparing transcriptomic differences on the skin of two ibises, between non-breeding and breeding state. They then try to understand the funtional relevance of regulated genes and infer their importance in the scope of ths trait development.

Overall I find that the conclusions are important but I think that the experimental design and intrinsic limitations of this methodology may not be sufficient for the level of inference that the authors state at the discussion. This could be easily resolved by some changes in the discussion.

First, two individuals may not be sufficient. In my experience, a number of five would be ideal, three would give some robustness to the findings. Meaning, that there is a necessity to assess if the results would be different if the sample size increased. For example, more diagnostic statistics to understand how similar were the responses of both ibises, could help in providing more robustness to your findings. In figure 3B we have this kind of diagnostics, but I wonder if this was done for all up-regulated genes.

Second, using an alternative patch of skin where this trait does not change between seasons would be important to understand which genes are upregulated in the cells under study in contrast to other skin cells. I understand that using other contrasts with internal tissues would be difficult in this species and in this experimental design but this could also have helped in some gene functions. Discussing whether this kind of contrast would help in the future could be interesting.

This issue is transversal to the whole study. Therefore I think that there is the need in the discussion of information about the limitations and how can we in the future increase the causality of the findings. What techniques could be used to infer the causality of the genes found to be important?

For example, within transcriptomic data sets, it is inevitable that the expressions of many genes will be highly correlated with each other and with outcomes of interest, and this makes it particularly difficult to separate association from causation in observational studies. Therefore, there is the need of complementary data to establish clear genotype-phenotype associations with a high standard of evidence. Are there additional experiments planned that will allow investigating the functional impact of these specific genes or mutations?

I also have some specific comments:

1. It is expected that feather condition should be at their best during breeding season. This could be done by an increase on feather condition related processes, namely the production of Uropygial secreted lipids. It is hard to discern wether this up-regulation in lipid metabolism is related to the production of these lipids or the "black substance". Here, using another skin patch would have helped cleraring this out. Please discuss this.

2. I could not understand if two males were used or the two gender?

3. How is the molt pattern? Did non-breding birds were sampled before, during or after the molt?

4. Immunity section in the discussion is too concise. Why do you think higher adaptative immune responses occured during the non-breeding season? Could there be a diversion of resources during breeding?

5. Discussion is too concise, as I told, and concluding remarks on the limitations and how can results be improved would increase the article robustness.

Table 2. At least one error on gene name TRY.

Round 2

Reviewer 1 Report

major problems are properly revised.